# Guided Imitation of Task and Motion Planning

**Michael J. McDonald**
University of California, Berkeley
m_j_mcdonald@berkeley.edu

**Dylan Hadfield-Menell**
Massachusetts Institute of Technology
dhm@csail.mit.edu

**Abstract:** While modern policy optimization methods can do complex manipulation from sensory data, they struggle on problems with extended time horizons and multiple sub-goals. On the other hand, task and motion planning (TAMP) methods scale to long horizons but they are computationally expensive and need to precisely track world state. We propose a method that draws on the strength of both methods: we train a policy to imitate a TAMP solver's output. This produces a feed-forward policy that can accomplish multi-step tasks from sensory data. First, we build an asynchronous distributed TAMP solver that can produce supervision data fast enough for imitation learning. Then, we propose a hierarchical policy architecture that lets us use partially trained control policies to speed up the TAMP solver. In robotic manipulation tasks with 7-DoF joint control, the partially trained policies reduce the time needed for planning by a factor of up to 2.6. Among these tasks, we can learn a policy that solves the RoboSuite 4-object pick-place task 88% of the time from object pose observations and a policy that solves the RoboDesk 9-goal benchmark 79% of the time from RGB images (averaged across the 9 disparate tasks).

## 1 Introduction

This paper describes a policy learning approach that leverages task-and-motion planning (TAMP) to train robot manipulation policies for long-horizon tasks. Modern policy learning techniques can solve robotic control tasks from complex sensory input [1, 2, 3], but that success has largely been limited to short-horizon tasks. It remains an open problem to learn policies that execute long sequences of manipulation actions [4, 5]. By contrast, TAMP methods readily solve problems that require dozens of abstract actions and satisfy complex geometric constraints in high-dimensional configuration spaces [6, 7]. However, their application is often limited to controlled settings because TAMP methods need to robustly track world state and budget time for planning [8, 9].

To address these concerns, two lines of work have emerged. The first uses machine learning to identify policies or heuristics that reduce planning time [10, 11, 12, 13, 14, 15]. This extends the domains where TAMP can be applied, but still relies on explicit state estimation. The second line of work learns policies that imitate the output of TAMP solvers. These approaches reduce dependence on state estimation by, e.g., learning predictors to ground the logical state [16] or training recurrent models to predict the feasibility of a task plan [17]. These learned policies operate directly on perceptual data, and do not need hand-coded state estimation. However, TAMP solvers define a highly non-linear mapping from observations to controls and this makes imitation difficult. Furthermore, the compute resources required for planning limits the availability of training data for learning.

In this paper, we build on both approaches. We take inspiration from guided expert imitation [18, 19] and train feed-forward policies to imitate TAMP. Our key insight is that Srivastava et al. [20]'s modular TAMP framework supports a distributed TAMP architecture that: 1) is optimized for throughput (as opposed to low latency, the typical focus in TAMP research); 2) trains policies to imitate the planner output; and 3) uses those policies to amortize planning across problems and reduce compute costs. The result is a virtuous cycle where learning accelerates planning, and faster planning provides more supervision. We propose a policy architecture that leverages the definition of the TAMP domain to improve learning performance. Task plans from our system supervise a task-level model that learns to predict a parameterized action. This output selects from and parameterizes a set of control networks, one for each action schema. We show that this architecture can execute manipu-

5th Conference on Robot Learning (CoRL 2021), London, UK.

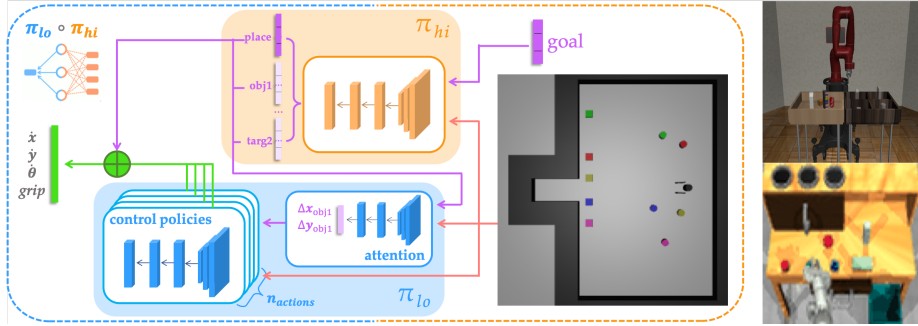

**Figure 1:** Left: We train hierarchical policies to imitate the output of a distributed task and motion planning system. The policy has two components, $\pi_{hi}$ and $\pi_{lo}$. $\pi_{hi}$ takes observations and a goal and produces a one-hot encoding of the choice of abstract action and associated parameters. These outputs and the original observation pass to the motion-level policy $\pi_{lo}$. $\pi_{lo}$ consists of two stages: an attention module and action-specific control networks. The attention module maps the continuous observation and discrete action parameters to a continuous parameterization. There is one control network per action type and the choice of abstract action gates the outputs of these controllers to produce the next control. We denote this combined policy as $\pi_{lo} \circ \pi_{hi}$. Right: We evaluate in two simulated robotics domains: RoboSuite [21] (top) and RoboDesk [22] (bottom). In RoboSuite we train policies from object poses that reach 88% success on the four object variant of the domain. In RoboDesk, we train policies from RGB image data and reach 79% success on the 9-goal multitask problem.

lation skills in high-dimensional environments from complex sensory input. As a result, our system is able to compile task and motion planning into a single feed-forward policy.

The contributions of this work are as follows: 1) we show a design for a distributed, asynchronous, high-throughput task and motion planning system that leverages policy learning to speed up planning; 2) we propose a hierarchical policy architecture that leverages the TAMP problem specification; and 3) we implement and evaluate this method to show it can learn to accomplish multiple goals over (comparatively) long horizons with high-dimensional sensory data and action spaces. In a 2D pick-place task, we train policies that place 3 objects precisely onto targets 83% of the time from RGB images. In the RoboSuite [21] pick-place benchmark we train policies for 7-DoF joint control that place 4 objects 88% of the time from object pose vectors. In the multitask RoboDesk benchmark, our learned policy averages a 79% success rate across 9 diverse tasks with 7-DoF joint control from RGB images.

## 2  Background

**Task and Motion Planning.** Task and motion planning (TAMP) divides a robot control problem into two components: a symbolic representation of actions (e.g., $grasp$) and a geometric encoding of the world. Task planning operates on a logical representation of the world. It finds sequences of abstract actions to accomplish a goal (e.g., pick($obj_1$), place($obj_1$, $targ_1$), ...). Each action encodes a motion problem that must be *refined* (i.e., solved) to obtain a feasible trajectory that satisfies specified constraints.

We represent TAMP problems with the formalism introduced by Hadfield-Menell et al. [23]. A TAMP problem is a tuple $\langle X, F, G, U, f, x_0, A \rangle$. $X$ is the space of valid world configurations. $F$ is a set of *fluents*, binary functions of the world state that characterize the task space $f : X \rightarrow \{0, 1\}$ (e.g. at($obj_3$, $targ_2$) or holding($obj_1$)). $G$ is the goal state, defined as a conjunction of fluents $\{g_i\}$. $U$ is the control space of the robot. $f$ describes the world dynamics: $f(x_t, u) = x_{t+1}$. $x_0$ describes the initial world configuration $x_0 \in X$. Finally, $A$ is the set of abstract *action schemas*. Each action schema $a$ has four components: 1) $a.params$: the parameters of the action (e.g., which object to grasp); 2) $a.pre$: a set of parameter-dependent fluents that defines the states when this action can be taken; 3) $a.mid$: a set of fluents that constrains the allowable controls for this action; and 4) $a.post$: a set of fluents that will be true after the action is executed. Solutions to such a problem are a pair of sequences $(\vec{a}, \vec{\tau})$, where $\vec{a}$ encodes abstract actions and $\vec{\tau}$ encodes refined motion trajectories. A plan is valid if 1) the initial state of each $\tau_i$ satisfies $a_i.pre$; 2) each $\tau_i$ satisfies $a_i.mid$; 3) the end state of each $\tau_i$ satisfies $a_i.post$; and 4) the final state satisfies the fluents that define the goal.

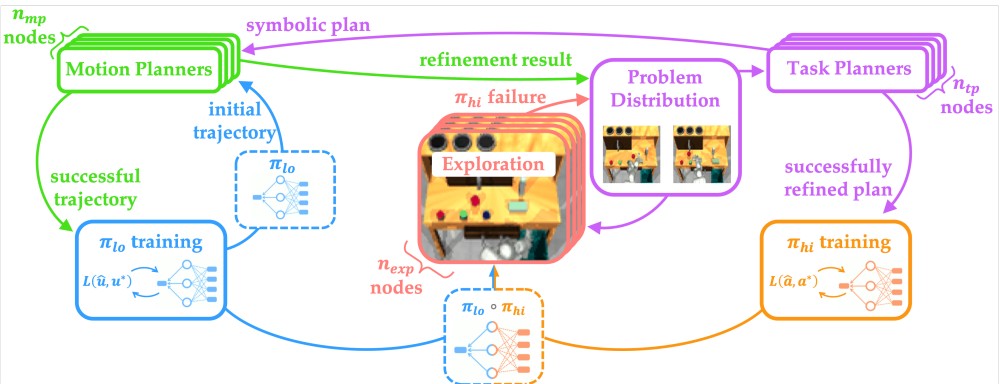

**Figure 2:** The system contains five core components which run in parallel and communicate through shared data structures. This division allows for the execution of arbitrary copies of motion planning, task planning, and policy rollouts and can scale to utilize all available hardware.

**Modular TAMP.** Our approach parallelizes the modular TAMP approach of Srivastava et al. [20]. They use a graph representation of TAMP, where nodes correspond to different task plans. A node can be either refined or expanded. Refinement uses motion planning to search for trajectories that accomplish the actions in the plan. Refinement failures are tracked along with the unsatisfiable constraints that caused the failure. When a node is expanded, one of these refinement failures is used to generate a new plan. The error information (e.g., a collision with an obstacle) is used to update the abstract domain so that the planner can identify an alternative plan (e.g., one that moves the obstruction out of the way). To solve a TAMP problem, Srivastava et al. [20] interleave refinement and expansion to find a plan that reaches the goal.

To implement plan refinement, we use the sequential convex optimization method described in Hadfield-Menell et al. [23]. This formulates motion planning as an optimization problem. The objective is a smoothing cost $\|\tau\|^2 = \sum_t \|\tau_{t+1} - \tau_t\|^2$. The constraints are determined by the pre, post, and mid conditions of an action schema: $\tau_0 \in a.pre$, $\tau_T \in a.post$, and $\tau \in a.mid$. E.g., for a grasp action the preconditions constrain the initial state to position the gripper near the object with proper orientation. The postconditions constrain the final state so that the object is in a valid grasp. The midconditions constrain the intermediate states to avoid collisions. If this optimization fails, we use the unsatisfied constraints to expand the associated plan node, as described above.

## 3   Method

This section has three parts. First, we propose a distributed TAMP solver. Then, we describe a hierarchical policy architecture that leverages the TAMP domain description to make it easier to model TAMP behavior. Finally, we show how to incorporate feedback from the learned policies to prevent trajectory drift.

### 3.1   Distributed Planning and Training Architecture

Figure 2 shows our overall design. Our system has four types of nodes that operate asynchronously: 1) policy training; 2) task planning; 3) motion planning; and 4) structured exploration. The task-level policy $\pi_{hi}$ and the motion-level policy $\pi_{lo}$ train within their respective nodes without backpropagation between the two. For this work we found standard supervised learning sufficient for good performance. However the modular design makes it straightforward to apply more complex procedures (e.g. generative adversarial imitation learning [24] or inverse reinforcement learning [25]). Specific network architectures and hyperparameters are included in the Supplemental Materials.

Algorithm 1 outlines our task planning procedure. It reads from a shared priority queue $Q_{task}$ that tracks task planning problems, represented as tuples of an initial state $x_0$, a logical state $\Phi_0$, a goal $g$, and a refined trajectory prefix $\tau_0$. At random intervals, or when the queue is empty, we sample a new planning problem from the base problem distribution, with an empty trajectory prefix $\tau_0 = \emptyset$. A symbolic planner, such as FastFoward [26], computes a valid action sequence $\vec{a}$ that achieves $g$ and pushes the problem $(x_0, \Phi_0, g, \tau_0)$ and action sequence $\vec{a}$ to the motion queue $Q_{motion}$.

Algorithm 2 outlines our motion planning procedure. It pulls from $Q_{motion}$ and applies the refinement procedure from section 2 to compute a valid trajectory segment $\tau^i$ for each $a^i$. If refinement fails, it computes the set of unsatisfiable constraints $\{e_i\}$ that describe why the motion planner could not refine $a^i$, appends these to the original logical description $\Phi_0$, and pushes the updated problem instance $(x_0, \Phi_0 \cup \{e_i\}, g, \tau^{0:i-1})$ to $Q_{task}$. If refinement succeeds, each $\tau^i$ is pushed to the motion dataset $D_{motion}$ to supervise the appropriate control policy. When the goal is reached (i.e., $\tau_T \in g$), it pushes the action sequence $\vec{a}$ to the task dataset $D_{task}$ to supervise the task-level policy.

## 3.2 Hierarchical Task and Motion Policies

Learning to imitate TAMP solutions with a single policy is difficult because small changes in the observation can lead to different task plans and, thus, radically different controls. We deal with this through a hierarchical policy architecture that mirrors the TAMP domain description. The policy is split into two parts: a task-level policy $\pi_{hi}$ to select parameterized actions $a_t$ and a motion-level policy $\pi_{lo}$ to select controls $u_t$ as a function of $a_t$ and world state. Figure 1 illustrates our design.

$\pi_{hi}$ maps continuous observations and goals to the discrete space of parameterized action schemas. We use the factored encoding scheme from Van et al. [12]. $\pi_{hi}$ outputs a sequence of vectors. The first specifies a one-hot encoding of the choice of action-type (e.g. place). Each subsequent vector provides a one-hot encoding of the ordered action parameters (e.g. $obj1$, $targ2$). Training data for this policy is pulled from the task dataset $D_{task}$.

The low-level policy $\pi_{lo}$ consists of two stages: an attention module and action-specific control networks. The attention module maps the continuous observation and discrete action parameters to a continuous parameterization. This step is flexible. In our system, it represents the relevant objects in a particular geometric frame. E.g., for the action $place(obj1, targ2)$ this is the pose of $obj1$ in the $targ2$ frame. When the policy has access to state data, we hard code this step. In other situations (e.g., learning from camera images), this is learned from supervision alongside the control policy. The abstract action, the continuous parameterization, and the original observation pass to the second stage to predict the next control. This stage contains a set of separate control networks, one per action schema. Training data for this policy is pulled from the motion dataset $D_{motion}$.

**Algorithm 1** Task Planning Node

**Require:** Shared queues $Q_{task}, Q_{motion}$
**Require:** Problem distribution $P$
 1: **while** not terminated **do**
 2:    **if** is_empty($Q_{task}$) **then**
 3:       $(x_0, \Phi, g, \tau) \sim P_{prob}$
 4:    **else**
 5:       $(x_0, \Phi, g, \tau) \leftarrow$ pop($Q_{task}$)
 6:    $\vec{a} \leftarrow$ task_plan($\Phi, g$)
 7:    push($Q_{motion}, (x_0, \Phi, g, \vec{a})$)

---

**Algorithm 2** Motion Planning Node

**Require:** Shared queues $Q_{motion}, Q_{task}$
**Require:** Expert datasets $D_{motion}, D_{task}$
**Require:** Motion policy $\pi_{motion}$
 1: **while** not terminated **do**
 2:    $(x, \Phi, g, \tau, \vec{a}) \leftarrow Q_{motion}$
 3:    **for** $a_i \in \vec{a}$ **do**
 4:       $\hat{\tau}^i \leftarrow$ rollout$(x, \pi_{lo}(*|a_i))$
 5:       $\tau^i$, success $\leftarrow$ motion_plan$(x, a_i, \hat{\tau}^i)$
 6:       **if** success **then**
 7:          $x \leftarrow \tau^i[-1]$
 8:          append($\tau, \tau^i$)
 9:          push($D_{motion}, (a_i, \tau^i)$)
10:       **else**
11:          ## add unsatisfiable constraints $\{e_i\}$
12:          push($Q_{task}, (x, \Phi \cup \{e_i\}, g, \tau)$)
13:          **break**
14:    **if** $\tau[-1] \in g$ **then** push($D_{task}, (\tau, \vec{a}, g)$)

## 3.3 Policy-Aware Supervision

A common challenge in imitation learning is that small deviations from training supervision build up over time [27]. To account for this, we propose task and motion supervision methods based on Dataset Aggregation [28]. For controls, we bias trajectory optimization to provide supervision near states encountered by the motion policies. After an initial training period, we warm-start optimization with a rollout from the appropriate motion policy. Then, to keep supervision close to this trajectory, we modify the optimization objective with a *rollout deviation cost*, based on the acceleration kernel from Dragan et al. [29]. This uses a generalization of Dynamic Movement Primitives [30] that adapts the rollout trajectory to satisfy optimization constraints. For a sampled trajectory $\hat{\tau}^i$, the optimization cost is $\|\hat{\tau}^i - \tau^i\|^2 = \sum_t \|(\tau^i_{t+1} - \tau^i_t) - (\hat{\tau^i_{t+1}} - \hat{\tau^i_t})\|^2$. This is conceptually similar to the regularization penalty from Guided Policy Search [18].

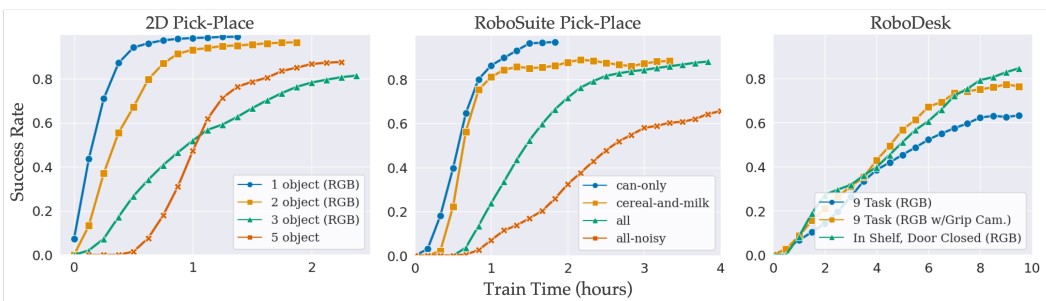

**Figure 3:** Average success rate over the course of training. Left: 2D pick-place domain. Training from RGB images and LIDAR-style sensors, the system learned to place 1, 2, and 3 objects 99%, 97%, and 83% of the time, respectively. Observing object positions, it learned to place 5 objects 88% of the time. Center: RoboSuite pick-place domain. Training from joint angles with object positions and orientations, the system learned to place 1, 2, and 4 objects 97%, 90%, and 88% of the time, respectively. With Gaussian observation noise it could place 4 objects 64% of the time. Right: RoboDesk domain. Training from joint angles and RGB images, the system learned to place the block in the shelf and close the door 89% of the time. Across the multitask 9-goal benchmark, the learned policy averaged a success rate of 68%, or 79% with an added gripper camera.

To generate task-level supervision, the exploration node identifies states where the task policy generates invalid actions. We sample a problem from the problem distribution and roll out the combined policy with an execution monitor that ensures that high-level action $a$ is only started when $a.pre$ are satisfied and is executed until $a.post$ are satisfied. We use this to build up a dataset of negative examples for the task policy and to identify states where TAMP supervision may improve performance. Details and pseudo-code are in the Supplemental Materials.

# 4 Experimental Results

We evaluate our system in three domains: a 2D pick-place simulator, the RoboSuite benchmark [21], and the RoboDesk benchmark [22]. Screenshots of the simulation environments are shown in Figure 1. We used FastForward [26] as the task planner. Unless otherwise stated, evaluations used 2 processes for task planning, 18 for motion planning, and, when applicable, 10 for supervised exploration. Experiments ran for 5 random seeds unless otherwise noted. Policy architecture details and hyperparameters are in the Supplemental Materials.

## 4.1 Learning 2D Pick-Place

**Setup.** We begin with a simple, synthetic, pick-place domain. The robot is circular with a parallel jaw gripper and its goal is to transfer up to 5 objects each to one of 8 randomly assigned targets. The domain has three action schemas: moveto-and-grasp($obj$), transfer($obj, targ$), place-and-retreat($obj, targ$). The full space includes a grounding of these for each object and target. With 5 objects and 8 targets, this gives 85 possible actions. The control space outputs x, y, and rotational velocities on the robot body and an open/close gripper signal. We used DMControl [31] for the underlying physics engine. The goal requires each object overlap with its target (i.e. $\|\text{pos}_{obj} - \text{pos}_{targ}\|_2 \leq \text{radius}_{obj}$). We use a simulated LIDAR sensor with 39 distance readings to facilitate collision avoidance.

**Manipulated Variables and Dependent Measures.** Our primary variable is the training procedure used. We compare against: 1) flat imitation learning (flat-IL), a single feedforward network trained to imitate TAMP output; 2) reinforcement learning with a dense reward function, trained with double the compute of our method (60 vCPUs); and 3) passive imitation learning (no-feedback), which uses the hierarchical architecture but turns off policy-aware supervision (20 vCPUs). We used two RL methods: PPO [32], a flat RL method, and HIRO [33], a hierarchical method. See the Supplemental Materials for training details. We used two performance measures: 1) success rate (Succ.), the rate at which trained policies achieve the goal; and 2) distance reduced (Dist), the percent reduction in the distance from each object to its target.

**Results.** Table 1 shows the results of these comparisons. With 1 object, we observed a 12% success rate for PPO and a 1% success rate for HIRO. Flat-IL succeeded 21% of the time in the 1 object variant. With two objects, it reliably reduced distance to the goal, but only succeeded 1.5% of the

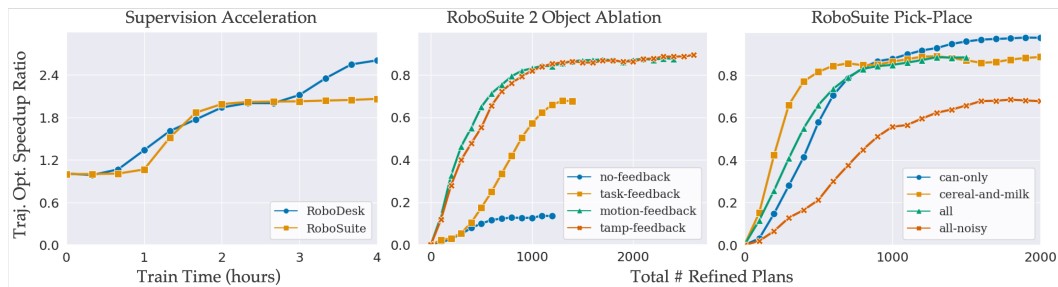

**Figure 4:** Our architecture utilizes partially trained motion policies to speedup plan refinement and improve data quality. Left: Speedup of plan refinement. During training, our method gave a 2x speedup of plan refinement in the RoboSuite domain with two objects and 2.6x speedup in the RoboDesk domain. Center: Ablation of performance as a function of supervision trajectories (Total # Refined Plans) for the RoboSuite domain with two objects. Right: Performance as a function of supervision trajectories across RoboSuite domains.

time. With 5 objects, we observed an 80% success rate for the hierarchical policy without feedback. This highlights the strong inductive bias our hierarchical architecture provides. With feedback, we observed success rates of 99.7%, 98.7%, and 88% for 1, 2, and 5 objects. Performance plateaued on the 5-object variant with approximately $3.6 \times 10^5$ environment transitions of supervision, generated by 1600 successful plans over 2 hours. See Figure 3 (left) for a depiction of performance over the course of training.

**Camera Observations.** We investigated the ability of our approach to train policies from RGB data. Figure 3 shows performance over time for solving problems with 1, 2, and 3 objects. After 4 hours of training (120 cpu hours) our policy successfully solves the 3 object task 83% of the time. In all cases, performance plateaued with less than $3.4 \times 10^5$ timesteps of supervision from the planner.

**Generalization.** We tested the learned policies' ability to generalize to novel goals and dynamics at test time. First, we introduced 6 additional obstacles to repre-

|          |       | 1 obj
3HL / 39TS |       | 2 obj
6HL / 85TS |       | 5 obj
30HL / 223TS |       |
|----------|-------|-------|-------|-------|-------|-------|
| Method   | Succ. | Dist. | Succ. | Dist. | Succ. | Dist. |
| PPO          | 12%   | 33%   | 0%    | 10%   | n/a   | n/a   |
| HIRO         | 1%    | 13%   | n/a   | n/a   | n/a   | n/a   |
| flat-IL      | 21%   | 55%   | 1.5%  | 51%   | n/a   | n/a   |
| no-feedback  | n/a   | n/a   | n/a   | n/a   | 80%   | 97%   |
| tamp-feedback| 99.7% | 99%   | 98.7% | 99%   | 88%   | 97%   |

**Table 1:** Comparison of average success rate (Succ.) and distance towards goal (Dist.) at completion of training for different learning methods on state data in the 2D Pick-Place domain. Dist. denotes the percentage reduction in the distance from each object to its target. HL denotes the number of high-level actions the planner would use to solve the problem and TS denotes the number of environment transitions induced by those actions.

sent 'humans' in the environment. The policy observations were the same as before, so the robot had to use the LIDAR sensor to avoid the humans. At training time the humans stayed in fixed, random locations. At test time, the humans use model predictive control to take actions that move towards a random goal location. Their objective penalized collisions with other humans, but not the robot. Rollouts terminated if a human collided with the robot. We observed a 50% success rate placing 3 objects in this condition. Next, we moved the target locations for objects from training to test time. We arranged the targets in a square at the center of the free space at training time. At test time, we moved the target locations to the edges of the space. This placed all targets outside the region seen during training. We observed success rates of 99%, 97%, and 39% for problems with 1, 2, and 4 objects, respectively.

## 4.2 Learning Pick-Place for 7-DoF Robotic Arm Control

**Setup.** We measured performance in the RoboSuite pick-place benchmark with a simulated 7-DoF Sawyer arm [21]. The goal is to move a cereal box, milk carton, soda can, and loaf of bread from random initial locations in a bin on the left to their target bins on the right. The world state is the joint configuration and the positions of the objects. The action space specifies velocities for the 7 joints and a binary open/close signal for the gripper. The symbolic domain has 16 high-level actions across 4 action schemas: moveto($obj$), grasp($obj$), moveholding($obj$), and putdown($obj$). The policy observations were joint angles, gripper state, the translation and rotational displacements from the gripper to each object, and the displacement from each object to its target. Our main experiment had 4 conditions: 1) can-only, to compare with prior work; 2) cereal-and-milk, the most

difficult 2 object condition; 3) all, the standard benchmark goal; and 4) all-noisy, which added Gaussian observation noise to the policy observations with standard deviation equal to $1\%$ of the possible deviation in initial values. We performed an ablation study in the cereal-and-milk condition to compare performance across feedback types: 1) no-feedback; 2) task-feedback (only); 3) motion-feedback (only); and 4) tamp-feedback (i.e., both feedback types).

**Dependent Measures.** To compare with published results in RL, we report the average dense return of the trained policies over 500-step horizons for can-only. Our primary measure of performance is success rate. We also measured the number of timesteps taken to reach the goal on successful trajectories. For the ablation study, we tracked the amount of time spent during motion optimization calls to measure the speedup from feedback.

**Comparison to Prior Work.** First, we compare against published results from RL and LfD. The official benchmark reports returns near 50 when using Soft-Actor Critic to move the can [21, 34]. In comparison, we observed an average return of 324 for this task. Fan et al. [35] report an average return below 600 over a 2000-step horizon. Our return for this horizon is at least 1734. We compare with LfD results on success rate. Mandlekar et al. [36] trained from the RoboTurk cans dataset [37] and observed success rates of 31% from state data and 43% from RGB camera images. In comparison, we observed a 98% success rate from state data for our can-only condition.

**Results.** Figure 3 (center) shows success rates over time for the four conditions described above. Our learned policies reach an 88% success rate in the all condition, 90% in the cereal-and-milk condition, and 98% in can-only condition. These correspond to average (successful) rollout lengths of 244, 120, and 55, respectively. In the all-noisy condition, we observed a success rate of $64\%$, compared with a 15% success rate executing plans from the TAMP solver directly. Figure 4 (center) shows success rate against training time for the ablation study. Both motion-feedback and tamp-feedback reach 80% success within 2 hours. Task-feedback shows steady improvement but does not reach the performance of the other variants within the time limit. (See the Supplemental Materials for a non-stationary condition where task-feedback improves on motion-feedback.) The no-feedback condition does not get above 20% success. We observe up to a 2x speedup of the motion optimization code with tamp-feedback compared to no-feedback. Figure 4 (left) illustrates the results.

**Generalization.** We ran a followup experiment to test the ability of the learned policies to generalize to unseen goals and dynamics. First, we held out goals that moved 3 and 4 objects and tested with other sizes of goals. When trained on problems that move 1 and 2 objects, the system was able to solve 3 object problems 48% of the time. When trained on 1, 2, and 4 object goals, it was able to solve 3 object problems 58% of the time. Next, we trained on 1, 2, and 3 object problems and observed a 60% success rate when tested with a goal to move all 4 objects. Note that this condition, which learns a multitask policy, is harder than the RoboSuite benchmark, which only attempts to learn a single-task policy.

Next, we used the domain randomization functionality of RoboSuite to test the ability of the previously trained policies to generalize to different dynamics. This varies properties of the physics engine such as the inertia and mass of the robot and objects, the parameters of the solver for contact forces, position and quaternion offsets of bodies, and physical properties of the joints. Using the default randomization parameters, we observed a success rate of 34%. When randomizing all parameters except the positional and quaternion parameters, we observed a 73% success rate.

### 4.3 Multitask Learning for 7-DoF Robotic Arm Control from RGB Images

**Background.** We completed the core of this work on May $19^{th}$, with the above domains [38]. Our final experiment anecdotally measures the engineering effort required to adapt the system to a previously unseen environment. The RoboDesk multitask benchmark had been released 6 days prior, on May $13^{th}$ [22]. Our final results reflect one engineer's effort over the course of three weeks to train on the benchmark, as well as two further weeks to train on custom variants of the benchmark. The primary engineering steps were: 1) extension of existing action schemas to handle new tasks; 2) integration of the existing system with the new simulator; and 3) iteration on motion specification and ML hyperparameters.

**Setup.** RoboDesk provides a 7-DoF PANDA arm with 9 disparate tasks: open slide, press button, lift block, open drawer, block in bin, block off table, lift ball, stack blocks, and block in shelf. We added a composite goal: block in shelf with door closed. Our policy observations contain only RGB

camera images and joint positions. The controls are on joint velocities and the gripper. Our domain has 7 action schemas: moveto($obj$), lift($obj$), stack($obj1, obj2$), place($obj, targ$), press($button$), open($door$), close($door$). This gives 31 possible high-level actions.

**Results.** Figure 3 shows training results over 10 hours on an Nvidia DGX Station (40 vCPUs, 4 GPUs). On the composite task, the policy had a success rate of 89%. On the benchmark with the default forward-facing camera, the learned policy had a success rate of 68% averaged across the 9 tasks. That rose to 79% when we added a second camera attached to the robot gripper. We report the benchmark performance for each goal in the Supplemental Material. In this domain, we observed a 2.6x reduction in motion optimization time from feedback as shown in Figure 4 (left).

## 5   Related Work

**Speeding up TAMP.** There is a growing field of work that uses machine learning on previous experience to reduce planning time for TAMP systems. One focus is on guiding the task-level search — something we do not explicitly address in our system. Chitnis et al. [15] trained linear heuristics from expert demonstrations. Kim et al. [11] used a score-space representation to guide the search by transferring knowledge from previous plans. Wells et al. [10] learned a classifier to assess motion feasibility and incorporated it as a heuristic into the search. At the motion level, Ichnowski et al. [13] also learned policies to warm-start trajectory optimization. The key differences are that they predicted full trajectories, rather than executing the policy, and they did not penalize optimization for deviating from the policy controls. We note that our focus is on the learned policies themselves, and so we do not directly compare the speedup produced by our system to other methods.

**Imitating TAMP.** Several other methods distill all or part of TAMP into learned policies. Paxton et al. [14] used imitation learning to train policies in an abstract task space and deep Q-learning in a control space. Like our system, they learned feed-forward policies for both task and motion prediction and used those policies to guide supervision. The key difference is the type of planner used. Ours is more compute intensive, but scales to larger problems. Kase et al. [16] proposed a method to imitate hand-engineered trajectories to train a model to predict logical state and controllers to execute actions. Their method does not directly predict tasks, but allows a TAMP system to run online. In contrast, we train a single policy that maps observations directly to controls.

The approach most similar to ours is that of Driess et al. [17]. They also train hierarchical policies to match TAMP output. They show that their system is able to integrate geometric and high-level reasoning for a reaching task that, depending on the position of the target, potentially requires the use of tools. The primary difference is that their approach predicts the feasibility of a given action sequence, based on the initial state, while we predict actions and controls given the current state. This means that they need to search over candidate high-level action sequences at test time. In our system, this search is handled by the task planning node during training and bypassed entirely at test time. The other large difference is that they represent their control policies with learned energy functions that are minimized online. Task transitions occur when this system reaches an equilibrium state. In contrast, we learn policies that directly choose when to transition between tasks.

## 6   Future Work

First, this system should be deployed on physical robots. We hope that our policies will transfer more readily than those produced by, e.g., RL because TAMP solutions do not rely on detailed dynamics models. Next, it would be interesting to experiment with parameter sharing for the motion policies to save space and potentially speed up training. It would also be interesting to experiment with the output space of the high-level policy to directly output continuous parameters, which may allow policies to generalize more effectively. There is likely room to improve on both the task and motion policy learning: sequence modeling methods from NLP [39] may be better suited to represent task plans and modern imitation learning procedures, such as GAIL [24] or related works, may reduce training time or improve generalization. Note that each of these changes is relatively straightforward to implement, as it only modifies a single node of the training architecture. Finally, it would be interesting to identify theoretical guarantees on the learned policies. For example, it may be possible to formalize this as optimizing policy parameters with respect to a mixed discrete-continuous cost function defined by the TAMP domain.

# 7 Acknowledgements

We wish to express our deepest appreciation to Professor Anca Dragan for her support and advice throughout the course of this work, and for providing the resources needed to see it through to completion. We would also like to deeply thank Thanard Kurutach for providing his advice throughout the project. We are furthermore grateful to our anonymous reviewers from the 2021 Conference on Robot Learning, whose feedback led to significant improvements both to the text and experimental results of this paper. This work was supported in part through funding from the Center for Human-Compatible AI.

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
