# OpenReview forum: "Guided Imitation of Task and Motion Planning"
_robot-learning.org/CoRL/2021/Conference — CoRL2021 Oral_

### Official Review · Reviewer_KfVB · 2021-07-22

**Originality:** Good
**Technical Quality:** Very Good
**Clarity Of Presentation:** Good
**Impact:** 4

**Recommendation:**

Weak Accept: I recommend accepting the paper, but will not argue for my recommendation if the majority of other reviewers have a different opinion.

**Summary:**

The authors propose a distributed training scheme for task and motion planning problems. This relies on access to symbolic robotics oriented task planner and motion planner to generate oracle training data for feed-forward policies that output task plans as action schemas and generate motion plans by chaining action schemas through reactive low-level control policies. The authors demonstrate the following:

1. Ammortizing planning by approximating TAMP solvers through hierarchical policy networks results in a high success rate for TAMP tasks.

2. Active exploration of problems space where the policy fails, and then generating oracle feedback can produce better quality data for optimization, by only using TAMP planner resources where feed-forward policies fail

3. Feedforward policies can speed up a planning based approach by generating better initial guesses.

**Issues:**

- Characterize the effect of planning horizon on the success rate. I would like to see how success rate deteriorates with longer required planning horizons

- A deeper discussion on differences between prior work on approximating TAMP solvers and how your approach builds on or is novel in that context.


**Reviewer Expertise:**

Very good: Comprehensive knowledge of the area

**Strengths And Weaknesses:**

**Strengths**

- The idea of ammortizing planning through approximation is an elegant way of retaining the flexibility of iterative planning, and online repair by narrowing the exploration space using good quality feed-forward policies. The authors idea of a distributed framework appears to be a powerful method to achieve this.

- The experiments are quite convincing towards the power of TAMP supervised feed-forward policies in comparison with RL-based or LfD-based policies. While this is to be expected as TAMP can provide a better training distribution, this work provides a convincing demonstration and a concrete modular implementation of this notion

**Methodological comments**

- The authors have restricted the action schemas to discrete outputs; the impact on problem domains that *require* continuous parameterizations in unclear. In theory,  the action selection network can output both discrete and continuous parameterizations, but the case of continuous parameters might be more challenging to train

- There are a lot of hand-engineered inputs provided to the problem (constraints, fluent descriptions etc.) that are not available to an RL-based baseline. While this is a representational advantage of using structured methods, this does place the burden of generating these on an automation engineer.

- The authors' decision to create a separate policy network for each action schema can potentially cause issues with more number of objects, shared parameters to reduce the memory requirements for control policy architectures would be interesting to look at.

- The authors do not demonstrate how the policy networks performance is impacted by length of planning horizon required to solve the test problems. Does performance deteriorate when longer policies and task plans are required?

**Presentation comments**

- What problems are picked by the exploration nodes? Are they picked from a procedural problem distribution, a dataset of problems? The main paper does not address the role of the exploration nodes at all

- Task planners are limited by the size of the logical states, the authors should include the number of logical fluents for each problem domain. Further, the authors do not provide any elaboration of how the geometrical facts for non-feasible task plans are generated.

- I would appreciate a longer discussion of how this work relates to prior approaches in learning to approximate TAMP solvers. I found this discussion quite superficial and focusing on the implementation details rather than the underlying aspects of TAMP solvers that are being learned by the models. In particular the approach by Driess et al. is closely related, and provides similar capabilities, but the authors' distributed approach can potentially provide faster scaling. The reader would appreciate a deeper comparison with prior work. The authors can potentially tighten the text in the results section or move some of it to the supplementary materials to provide a better framing of their approach.

**Summary Of Recommendation:**

I believe that the idea of decentralized learning for approximating a TAMP solver is novel, yet the idea of approximating the TAMP solver by itself is not. I believe that the authors should frame their contribution as such

Further, the larger ideas are quite clear in the paper, but there are a lot of implementation details that are not elaborated enough.

I don't believe the work is purely incremental.

Therefore overall I recommend a weak accept, and I urge the authors to make some changes

---

> ### Author Response · Authors · 2021-08-30
> **Response to Reviewer KfVB**
>
> Thank you for your detailed comments. We have updated the future work section to address the methodological comments. We answer in detail below:
>
> - We expect the high-level policy to work well, if not better, with continuous outputs. We selected to run our experiments with discrete outputs to highlight the flexibility of the types of high-level state that we can incorporate into the hierarchy.
>
> - We agree that TAMP systems do require additional information beyond an RL formulation. However, we believe this is easier to manage than tuning hyper-parameters in RL. Consider, for example, our experiments applying HiRO, a hierarchical RL approach, to the 2D pick-place domain with one object. It is hard to identify a good goal space to use for this problem, and even more difficult to figure out how to generalize this to the problems with multiple objects. In comparison, we were able to adapt our system to a previously unreleased multi-task robot domain with high-dimensional observations with minimal modifications to our existing hierarchy and codebase.
>
> - We agree that it would be preferable to do parameter sharing for the control policies. We expect that these approaches would achieve similar performance and have added this to the future work section. In earlier experiments, we used a single control policy and observed similar results. We opted for the current implementation for code simplicity. We would like to point out, however, that the number of policies needed depends on the number of actions in the planning domain, which is independent of the number of objects in the domain. We have clarified this in the paper.
>
> - We did not directly explore the impact of plan length on the policy performance. In the course of running our new goal generalization experiments in RoboSuite, we did however vary the number of objects to move to their target locations. We found that the system had an average success rate of 80% with 1 object, 78% with 2 objects, 73% with 3 objects, and 60% with 4 objects when trained from problems with 1 to 3 objects. These correspond to average plan lengths of 60, 125, 230, and 260 environment transitions (we only check for success at intervals of 20 timesteps, hence the clean averages). We have updated the paper to include this in the results section.
>
> We have updated the paper to address the presentation comments. Problems for the exploration node are sampled from the same problem distribution as the planning problems. We did not optimize the domain descriptions, as symbolic planning was fast in comparison to the other issues considered. As a result, the number of fluents (which is 87) is not representative of the domain size, as it, e.g., contains support for bimanual manipulation domains used in different applications of our approach. We have updated the domain descriptions to include information about the size of the domains.
>
> We have also expanded on the comparison to previous work to discuss the difference between our approach and that proposed by Driess et al. The updated paragraph is included below for convenience:
>
> The approach most similar to ours is that of Driess et al. They also train hierarchical policies to match TAMP output. They show that their system is able to integrate geometric and high-level reasoning for a reaching task that, depending on the position of the target, potentially requires the use of tools. The primary difference is that their approach predicts the feasibility of a given action sequence, based on the initial state, while we predict actions and controls given the current state. This means that they need to search over candidate high-level action sequences at test time. In our system, this search is handled by the task planning node during training (and bypassed entirely at test time). The other large difference is that they represent their control policies with learned energy functions that are minimized online. Task transitions occur when this system reaches an equilibrium state. In contrast, we learn policies that directly choose when to transition between tasks.

---

### Official Review · Reviewer_1SbR · 2021-07-23

**Originality:** Excellent
**Technical Quality:** Very Good
**Clarity Of Presentation:** Excellent
**Impact:** 4

**Recommendation:**

Strong Accept: I recommend accepting the paper and will argue for my recommendation even if other reviewers hold a different opinion.

**Summary:**

This paper presents a framework for learning hierarchical long-horizon policies using a task and motion planner in the loop. The framework is based on a flexible TAMP algorithm, which uses an explicit interface between task planning and motion planning. Using this decoupled approach allows performing each part independently and in parallel. The task planner uses a Fast Forward algorithm to find sequences of actions, while motion planning is performed using numerical optimization.

Since the output of the framework is a trained neural network policy, the TAMP plans are not necessarily carried out completely, rather they are used to generate datasets to train two policies (i.e., task and motion). These are in turn used to guide the search.

Hence the authors propose feedback mechanisms between the hierarchical policies and the TAMP algorithm which provide the training dataset for the policies. These feedback loops allow lowering the sample complexity of the framework. The proposed approach is thoroughly tested using three different benchmarks using ablation as way to highlight the important part of the framework.


**Issues:**

See above.

**Reviewer Expertise:**

Very good: Comprehensive knowledge of the area

**Strengths And Weaknesses:**

The paper is well written and presents the ideas and results in clear way. The main strength of the paper is twofold.

1)	The originality of the idea is very relevant and design choices are sound
2)	The choice experiments are interested and show the potential of the framework

Efficient motion policies for long-horizon manipulation tasks is clearly a very important milestone for robotics research. In this context very, few works combine learning and planning in an original way. As the authors point out most of the attempts use learning in the planning algorithms. In this work, the authors intertwine the two in such way that it minimizes sample complexity. For that using [15] seems very appropriate as it allows to decouple the task and motion planning problem.

Having experiments on three benchmarks allows to see, which parts of the framework are more important under which condition. For instance, considering feedback loops, the motion-feedback seems most important but sometimes task feedback can improve sample complexity in domains where the planner does not encounter the situations produced by the policy.

While these are two strong points of the paper. They could be further improved by considering the following: The paper mainly lacks a theoretical situation of the problem.

While it seems difficult to analyze theoretically this work, the framing of the research is not completely accurate. The title suggests that the authors are solely imitating the TAMP policies and that they use feedback to increase sample complexity. While that may be true in practice, it seems that a more accurate framing of the work is terms of Reinforcement Learning. The work can be viewed as a hierarchical version of Guided Policy Search (GPS). This way it should be possible to provide a better theoretical derivation and experimental analysis of the work.

For instance, taking inspiration from GPS, one could study the trade-off for generating policies that suddenly diverge from what the TAMP algorithm produces. This could be enforced in the motion optimization component using a cost function as done in GPS rather than only from rollout initialization. A dynamics model could be learned for the motion optimization, a KL constraint could be added in the supervised training phase etc.

Concerning the theory, deriving the approach as some kind of descent algorithm would provide more clarity even if the implementation approximates it.

Finally, concerning the experiments framing the problem as a model-based hierarchical RL algorithm naturally yields comparison to other hierarchical RL approaches such as HIRO, FuN etc. which are absent from the paper.


**Summary Of Recommendation:**

It is recommended to frame this work in the context of Model-based hierarchical Reinforcement Learning, instead of active imitation learning. This reframing implies comparing to SOTA HRL algorithms such as HIRO or FuN.

---

> ### Author Response · Authors · 2021-08-30
> **Response to Reviewer 1SbR**
>
> Thank you for your comments. We agree that the paper would benefit from stronger theoretical guarantees, however our primary goal was to build a system that could effectively learn to execute long-horizon tasks from perceptual data. We agree about the close connection to GPS, in fact that is what initially inspired our approach. We have updated the paper to better convey this connection. While we agree that it would be nice to do a more faithful reproduction of GPS’s learned cost functions and KL constraints, early experiments showed that they performed poorly, which led to the version of the system that we presented. It would be interesting to revisit this with the current results in hand to develop an approach that more faithfully implements GPS. We have updated the future work section to suggest this to readers.
>
> We also agree that this work should be considered in relation to model-based hierarchical reinforcement learning. We have run experiments in the 1 object 2D pick-place domain with the h-baselines implementation of HiRO, using TD3 as the underlying RL algorithms. We are continuing to experiment with different hyper-parameters, however we have not yet been able to find a configuration that consistently achieves the goal. Our experiments have used 30 processes with 6 hours of run time. We believe this illustrates the benefits of the explicit hierarchy encoded in the TAMP formalism. With feudal-style RL, the euclidean distances used to measure distance-to-goal are not great proxies for the actual value function. Small changes in the commanded goal can lead the robot to miss the object or to knock it further away from itself and its goal. With some careful reward design, we were able to improve the performance of PPO to 12%, but we have not been able to move beyond this. We have updated the paper to reflect this discussion.

---

> > ### Comment · Reviewer_1SbR · 2021-09-02
> > **Comment to Rebutal**
> >
> > Thanks for taking into account the review. I think this improved the paper.

---

### Official Review · Reviewer_i7Gj · 2021-07-25

**Originality:** Good
**Technical Quality:** Good
**Clarity Of Presentation:** Very Good
**Impact:** 4

**Recommendation:**

Strong Accept: I recommend accepting the paper and will argue for my recommendation even if other reviewers hold a different opinion.

**Summary:**

The authors propose an imitation learning method for long-horizon robot tasks that is train on task and motion planning (TAMP) data. They train a hierarchical policy, with a high-level policy that learns to predict abstract/symbolic actions and a low-level policy which has an attention component and generates low-level motions. The goal is to generate a sequence of actions that both (1) generates high-level symbolic plans, and (2) generates trajectories.

Training is done via a system inspired by Dagger where they run a task and motion planner in parallel alongside training. After training, they run plans and if they fail, collect additional training examples by invoking the expert TAMP system.

Training runs as a set of nodes: the "task planning" node runs something like FastForward to generate a sequence of high-level actions, a motion planning module which generates trajectories, and training. A motion planning node generates trajectories, a training node updates the model, and finally they perform structured exploration to collect more data (similar to Dagger) to do both. This is the big innovation the authors propose, which allows them to efficiently train this hierarchical policy on a variety of simulated tasks.

They perform experiments on 3 simulated domains, 2 of which are from RGB images.

**Issues:**


My big concern with this method is that it seems extremely data intensive and not very practical. The system as a whole seems to work on environments and tasks that have been seen; one of the big advantages of TAMP systems though is generalization to new problems.

How much data did your system end up needing? How many trajectories were collected? Some additional information would help determine how well it scales.

Ablation experiments holding out specific types of goals in RoboSuite would be very interesting and helpful.

Please make plots like Fig. 5 in color as well as using symbols; different shades of gray are very hard to parse.

Minor:
Abstract: let's --> lets
3.1:  A symbolic planner, such as Fast123 Fowardciteff, --> FastForward~\cite{...}


**Reviewer Expertise:**

Excellent: Expert knowledge on the topic of the paper

**Strengths And Weaknesses:**

Briefly:
+ Important problem
+ Reasonable performance on common benchmarks
+ Approach for training seems reasonable, and the distributed training architecture seems novel and helpful for large tasks where simulation is an option
- numerous obvious typos
- state space often not clear to me
+ Good comparisons to related work

I think the chief novelty is their distributed training architecture. The authors propose a system which shows performance on a range of different environments. They also show results both on RGB images and on joint state vectors and on a variety of problems and environments.

I have some concerns about how this system would scale, though. The authors performed some ablation experiments were performed; however, these weren't performed in the cases where the system was trained on RGB images.

The meaning of the object ablation experiments in RoboSuite seemed unclear to me; the authors should explain this in more detail and what it says. They do an analysis of different types of feedback, which is interesting. It seems as if tamp-feedback is not useful more than motion-feedback, which might imply the high-level task structure is too simple.


**Summary Of Recommendation:**

I think this paper tries to solve an important problem and brings some interesting ideas, and they get decent results, even if the results are not demonstrated in the real world and the generalization experiments they perform are overly limited. The ablation experiments focus on feedback types, not on holding out different goals.

The distributed training system seems like it could be very helpful, though, if it could be applied to training models that could generalize or adapt to real problems.

UPDATE: after the author's response I think it's more practical and stronger results than I had initially thought. Updated.

---

> ### Author Response · Authors · 2021-08-30
> **Response to Reviewer i7Gj**
>
> Thank you for your feedback. We have corrected the typos that you identified and switched to figures that use color to reference different data series.
>
> To address the comment on generalization, we would like to highlight several of our experiments that assess generalization. The first is the non-stationary environment (l.219) where the policies were trained with stationary obstacles and then operated at test time with moving obstacles whose dynamics were defined by MPC for 6 ‘humans’ moving towards a target. These actors avoided collisions with each other, but not the robot. We had a 50% success rate with 3-object goals. In addition, we have run a set of new experiments that we are working to incorporate into the paper, while respecting the page limits. We summarize the results here.
>
> First, we ran an additional set of experiments in the 2D pick-place environment. We arranged the targets in a square at training time and tested on a larger square so that all targets were outside the region encountered during training. We ran experiments with 1, 2, and 4 objects and observed success rates of 99%, 97% and 39%, respectively. These correspond to a >99%, 98%, and 63% average reduction in the distance of each object to its target.
>
> Second, we ran goal generalization experiments in the RoboSuite domain, as requested. We performed 4 different variants of this experiment: 1) we held out 3 object tuples of goals and trained on all other subsets; 2) we held out 3 object tuples of goals and trained on 1 and 2 object goals; 3) we held out the 4 object goal and trained on 1, 2, and 3 object goals; and 4) we repeated (3) where distractor objects were initialized at their respective goals. We observed the following results: 1) 56% success rate, 38% avg reduction in distance to goal; 2) 48% success rate, 34% avg reduction in distance to goal; 3) 49% success rate, 60% avg reduction in distance to goal; 4) 60% success rate, 67% avg reduction in distance to goal.
>
> We believe that this shows two distinct ways our system can generalize to new goals at test time. Furthermore, we have also added experiments with dynamics generalization to our results in the RoboSuite domain. We used the framework’s existing domain randomization wrapper, which alters properties of the environment such as mass and inertia, solver parameters for contact forces, physical joint properties, position and orientation offsets, etc. With the default settings, the previously trained policies from our system had a success rate of 34%. When we disabled the randomization to position and orientation (but left all other settings at their default values), that success rate rose to 73%.
>
> We apologize that we did not provide more information on the amount of data that our system used to train. We have updated the presentation of the result to include the performance of the RoboSuite policy as a function of the number of plans generated. Our success rate plateaued after solving 1900 problems to train the one object task, 1300 problems to train the 2 object task, 1200 problems to train the 4 object task and 1800 problems to train the noisy variant of the 4 object task.  In the 2D pick-place experiments, our success rate plateaued after solving 1600 plans for the 5 object variant, 2200 plans for the 3 object RGB experiment, 4000 for the 2 object RGB experiment, and 6000 for the 1 object RGB experiment. The increase in the number of plans solved for simpler problems occurs because the simpler problems are easier to solve (so the planner has higher throughput) and provide less supervision per plan.
>
> We also like to highlight that our results were all done on a single machine using 30 virtual cpus and 4 hours real time. We believe this is a substantial reduction on the compute cost required to train, e.g., vision-based policies for robot control. Furthermore, because the behaviors generated by our system are able to generalize across robot dynamics (even when not trained to do so), we believe that it will be possible to adapt these policies to real robots with minimal modifications.
>
> Finally, we would like to emphasize that, to our knowledge, our performance on RoboSuite and RoboDesk is state-of-the-art. We were unable to find policy learning approaches that scaled beyond 1 object for RoboSuite (either reinforcement learning or learning from demonstrations). The RoboDesk benchmark is too new to have published results, however, we would like to emphasize that we successfully trained a multi-task policy to control a simulated robot arm from pixels. The domain was designed, according to its release notes to “contain objects of different shapes and colors, whose initial positions are randomized to avoid naive memorization and require learning algorithms to generalize.”

---

> > ### Comment · Reviewer_i7Gj · 2021-09-03
> > **Final response**
> >
> > Thanks for your updates, I do think they improve the paper.
> >
> > > we would like to emphasize that we successfully trained a multi-task policy to control a simulated robot arm from pixels.
> >
> > This is actually not very clear from the paper! Please revise it in the final version to make your world state clear. It looked like it was object positions.
> >
> > I think that state-of-the-art performance on robodesk and robosuite is great. The authors could stand to improve clarity of model inputs and architecture especially. But I think this is a very interesting paper that has a bunch of great ideas in it, so I'll upgrade my review to a Strong Accept.

---

### Meta-Review · Area_Chair_GR3P · 2021-08-14

**Recommendation:** Accept (Oral)
**Confidence:** 4

**Metareview:**

This paper addresses the issue of jointly optimizing task and motion planning (TAMP), and proposed a method which includes to imitate a TAMP solver with a trained policy and to fast update the TAMP solver with a hierarchical policy architecture.

I agree with reviewers that this paper is well organized and the idea is quite novel, the experimental results illustrated this sutdy is effective and promising. I believe this is a good work.

I also agree with reviewers on some minor points they mentioned, and suggest authors carefully address these comments to further improve this paper.

More added:
I would like to thank authors for your carefully and detailed responses to comments raised by reviewers. Authors responses seems to be effective, since the score of this paper was improved after rebuttle.
Also thanks feedbacks from reviewers.
I really believe the revised version is more better.
Congratulations!

---

### Decision · Program_Chairs · 2021-09-13

**Decision:**

Accept (Oral)

**Comment:**

This paper addresses the issue of jointly optimizing task and motion planning (TAMP), and proposed a method which includes to imitate a TAMP solver with a trained policy and to fast update the TAMP solver with a hierarchical policy architecture.

I agree with reviewers that this paper is well organized and the idea is quite novel, the experimental results illustrated this sutdy is effective and promising. I believe this is a good work.

I also agree with reviewers on some minor points they mentioned, and suggest authors carefully address these comments to further improve this paper.

More added:
I would like to thank authors for your carefully and detailed responses to comments raised by reviewers. Authors responses seems to be effective, since the score of this paper was improved after rebuttle.
Also thanks feedbacks from reviewers.
I really believe the revised version is more better.
Congratulations!